## RESEARCH ARTICLE

# *Drosophila* and human cell studies reveal a conserved role for CEBPZ, NOC2L and NOC3L in rRNA processing and tumorigenesis

Guglielmo Rambaldelli[1], Valeria Manara[2], Andrea Vutera Cuda[2], Giovanni Bertalot[2,3,4], Marianna Penzo[1,5] and Paola Bellosta[2,6,*]

## ABSTRACT

NOC1, NOC2 and NOC3 are evolutionarily conserved nucleolar proteins that play an essential role in the maturation and processing of ribosomal RNA (rRNA). NOC1 in *Drosophila* is necessary to sustain rRNA processing, whereas its depletion leads to impaired polysome formation, reduced protein synthesis and induces apoptosis. In this study, we demonstrated that the RNA-regulatory functions of NOC1 are conserved in vertebrates, where the reduction of CEBPZ, the homolog of NOC1, leads to the accumulation of unprocessed 45S pre-rRNA, a reduction in protein synthesis, and inhibition of cell growth. Gene Ontology and bioinformatic analyses of CEBPZ, NOC2L and NOC3L in tumors highlight a significant correlation between their expression and processes that regulate rRNA processing and ribosomal maturation. Moreover, comparative analysis of TCGA datasets from tumor databases revealed that CEBPZ, NOC2L and NOC3L exhibit contrasting expression patterns across tumor types. This context-dependent behavior suggests that overexpression of these proteins might promote tumor growth, whereas reduced expression could exert tumor-suppressive effects, underscoring their complex and unexpected regulatory roles in cancer.

KEY WORDS: NOC proteins, CEBPZ, NOC2L, NOC3L, Ribosome biogenesis, rRNA processing, Nucleolar proteins, Cancer, Evolutionary conservation, *Drosophila*

## INTRODUCTION

Ribosome biogenesis is a highly regulated process that relies on specific nucleolar proteins to facilitate the processing of ribosomal RNA (rRNA), which is essential for proper ribosome maturation. Disruptions in this pathway can result in aberrant ribogenesis and contribute to proliferative disorders, including tumor initiation

[1]Department of Medical and Surgical Sciences, University of Bologna, Via Massarenti 9, 40138 Bologna, Italy. [2]Department of Cellular, Computational and Integrative Biology (CIBIO), University of Trento, Via Sommarive 9, 38123 Trento, Italy. [3]Unità Operativa Multizonale di Anatomia Patologica, APSS, 38123 Trento, Italy. [4]CISMed, University of Trento, Via Santa Maria Maddalena 1, 38122 Trento, Italy. [5]IRCCS Azienda Ospedaliero-Universitaria di Bologna, 40138 Bologna, Italy. [6]Department of Medicine, NYU Langone School of Medicine, 550 First Avenue, New York, 10016 NY, USA.

*Author for correspondence (paola.bellosta@unitn.it)

G.R., 0009-0000-5650-2892; V.M., 0000-0001-9292-6588; A.V.C., 0009-0006-8196-9455; G.B., 0000-0002-4862-7705; M.P., 0000-0002-1738-9767; P.B., 0000-0003-1913-5661

(Dorner et al., 2023; Hwang and Denicourt, 2024; Penzo et al., 2019). Nucleolar Complex Proteins 1, 2 and 3 (NOC1, NOC2 and NOC3) are evolutionarily conserved nucleolar proteins essential for growth in *Saccharomyces cerevisiae* and *Arabidopsis* (Edskes et al., 1998; Milkereit et al., 2001; Li et al., 2009). Their function was first characterized in yeast for their ability to form functional NOC1–NOC2 and NOC2–NOC3 heterodimers, which are necessary for the processing and transport of the rRNA during the maturation of the 60S ribosomal subunit (Milkereit et al., 2001). In addition, NOC proteins are crucial for maintaining nucleolar integrity and facilitating the incorporation of ribosomal proteins into ribosomes (Dorner et al., 2023; Hurt et al., 2024; Sanghai et al., 2023; Vanden Broeck and Klinge, 2023), ensuring proper ribosome biogenesis. NOC proteins are also present in *Drosophila*, where we have demonstrated their essential role in controlling organ and animal growth (Destefanis et al., 2022). Furthermore, we have shown that NOC1 controls rRNA processing, and its reduction results in rRNA accumulation, disrupts polysome assembly and compromises protein synthesis (Destefanis et al., 2022). Additionally, mass spectrometry analysis of the NOC1 interactome identified NOC2 and NOC3 among the interacting proteins, suggesting that NOC1 might also form heterodimers with NOC2 and NOC3 in *Drosophila*, as observed in yeast (Manara et al., 2023).

In humans, the homologs of NOC1, NOC2 and NOC3, called CEBPZ, NOC2L and NOC3L, respectively, are essential and in mice, their respective gene knockouts are embryonic lethal (Friedel et al., 2007). Additionally, data from DepMap, a platform that reports genetic dependencies in cancer (Fong et al., 2025; Meyers et al., 2017; Tsherniak et al., 2017), indicates that they are all essential for tumor growth (Ciani et al., 2022).

CEBPZ (ID10153) belongs to the C/EBP (CCAAT protein) family of transcription factors (Lum et al., 1990). Known for its role as a DNA-binding transcriptional activator of heat shock protein 70, the function of CEBPZ in vertebrates remains largely unexplored. In tumors, CEBPZ has been proposed as a marker for colorectal and gastric cancers, whereas point and missense mutations have been identified in individuals with acute myeloid leukemia (AML) (Herold et al., 2014), where CEBPZ works as a cofactor to maintain m6A modifications in leukemia cells (Barbieri et al., 2017).

NOC2L (also known as NIR, inactivator of histone acetylases) is a nucleolar histone acetyltransferase inhibitor that represses the transcription of genes controlling the cell cycle (Li et al., 2021). Conditional knockout mice exhibit increased p53 levels, apoptosis and bone marrow defects (Ma et al., 2014). Interestingly, silencing of NOC2L results in pre-rRNA accumulation and increases p53 levels (Wu et al., 2012). Depletion of NOC2L in LOVO tumor cells inhibits their growth in xenograft mice (Li et al., 2021), supporting a role in tumor progression. Moreover, elevated NIR levels in colorectal

cancer serve as a marker for malignancy in individuals with poor disease outcomes (Li et al., 2021).

NOC3L (also known as Fad24, factor for adipocyte differentiation 24), located in the nucleolus, is required for the adipocyte differentiation of NIH-3T3-L1 cells (Tominaga et al., 2004). Whereas its overexpression is linked to colorectal and gastric carcinomas, its reduction suppresses the *in vitro* growth of gastric cancer cells (Yan et al., 2020), suggesting a similar behavior to that reported for NOC2L in LOVO cells (Li et al., 2021). Notably, in zebrafish, noc3l reduction significantly affects the development of hematopoietic cells (Walters et al., 2009), mirroring our unpublished observation on CEBPZ reduction in zebrafish, suggesting a role in the differentiation of hematopoietic cells. Interestingly, NOC3, known as Noc3p in yeast, was found to be essential for the control of the DNA replication initiation complex (Zhang et al., 2002). This function was later confirmed for its human homolog, NOC3L, which supports DNA replication and cell proliferation and also contributes to pre-initiation complex (PIC) formation (Cheung et al., 2019), suggesting additional moonlighting activities for these proteins.

This study uncovers a conserved and crucial function for the NOC proteins in ribosomal biogenesis. We show that in *Drosophila*, as for yeast, NOC1, NOC2 and NOC3 are indispensable for properly processing rRNA. We also propose that CEBPZ, NOC2L and NOC3L collaboratively regulate rRNA maturation in humans, and that disruption of any of these leads to defective rRNA processing, impaired ribosome assembly and cell death. GO analysis revealed that CEBPZ, NOC2L and NOC3L are associated with RNA processing and rRNA maturation. Notably, co-expression analysis of CEBPZ and NOC2L showed a significant enrichment in genes that participate in R-loop metabolism. These findings support a recent study identifying CEBPZ as part of the protein complexes associated with R-loop resolution in human stem cells (Wu et al., 2021). R-loops are structures of RNA–DNA hybrids normally formed during transcription, also present in the nucleolus due to the high rate of transcription of rRNAs (Feng and Manley, 2022; Petermann et al., 2022; Wells et al., 2019).

Expression profiling across diverse tumor types revealed distinct patterns for CEBPZ, NOC2L and NOC3L, with them being elevated in several cancers, yet markedly reduced in AML and kidney chromophobe carcinoma (KICH). This distinctive and intriguing pattern correlates with poor prognosis in specific cancer types and suggests a context-dependent, cell-type-specific role for these proteins in tumor progression. Notably, this variation might be related to their potential involvement in regulating R-loops, pointing to a broader role in maintaining genomic stability beyond ribosome biogenesis.

## RESULTS
### Loss of NOC proteins disrupts rRNA processing, elevates p53, and impairs growth in *Drosophila* and human cells

Ubiquitous reduction of NOC1, NOC2 or NOC3 in *Drosophila* impairs animal development, with NOC1 reduction affecting protein synthesis and inducing apoptosis (Destefanis et al., 2022). Here, we show that this effect is common for all three NOCs, as reduction of NOC1, NOC2 or NOC3 affects the correct maturation of rRNAs, as shown by the accumulation of *ITS1* and *ITS2* pre-rRNA forms (Fig. 1A,B) and by reduction of the mature *18S* and *28S* ribosomal RNAs (Fig. 1C,D). In addition, we also found upregulation of p53 at transcriptional levels (Fig. 1E), supporting the increase in apoptosis observed previously, as a response to stress (Destefanis et al., 2022).

This mechanism appears to be conserved in humans, as reducing levels of the human homolog CEBPZ in HEK293T tumor cell line

induces the accumulation of *45S* pre-RNA and increases the levels of *p53* (also known as *TP53*) mRNA, and its target *Bcl2* and *Bax* (Fig. 1F). Similar results were observed in the HepG2, liver carcinoma cell line, where CEBPZ reduction also leads to increased levels of *45S* pre-RNA (Fig. 1G) and elevated p53 protein level (Fig. 1H). A SUnSET assay showed that there was a significant reduction in puromycin incorporation in cells silenced for CEBPZ, indicating decreased protein synthesis (Fig. 1I,J). This reduction correlates with impaired cell growth, as shown by a clonogenic assay (Fig. 1K–M). These data recapitulate the observations in *Drosophila* and support the hypothesis that CEBPZ plays an evolutionarily conserved role in regulating rRNA maturation.

Human NOC proteins share 32% amino acid identity with their *Drosophila* homologs (Fig. 1N). Analysis to predict their 3D structures using AlphaFold (Jumper et al., 2021; Varadi et al., 2022) imported into ChimeraX (Goddard et al., 2018; Meng et al., 2023) revealed the presence of highly similar structures in the proteins of the two species, with 37%, 47% and 40% of identity for CEBPZ, NOC2L and NOC3L, respectively (Fig. 1O). Both CEBPZ and NOC1 contain a CAAT-binding protein (CBP) and NOC domains, the latter of which is also present in NOC3L and NOC3 (Fig. 1N,O).

Based on these conserved structural similarities, we hypothesized that CEBPZ might be part of the complex in which heterodimers with NOC2L participate in the transport and cleavage of rRNA and maturation of the 60S ribosomal subunit, and a model has been created to explain these passages (Fig. 1P) (Hurt et al., 2024). During these maturation events, CEBPZ–NOC2L heterodimers are formed and, in the state A (Fig. 1P), likely interact with proteins including PES1, BOP1, FTSJ3 and WDR1, as determined by a Gene Ontology (GO) analysis that showed their gene expression was significantly correlated with expression of CEBPZ–NOC2L (Table 1; Fig. 1P). In further maturation steps, CEBPZ detaches from NOC2L, allowing NOC3L to bind and heterodimerize with NOC2L in a complex necessary at state C of the 60S ribosomal maturation (Hurt et al., 2024). In addition to FTSJ3 and PES1, this complex likely contains NIP7 and DIS3, expression of which were found to be coregulated with NOC2L–NOC3L in our analysis (Table 1; Fig. 1P).

### Gene Ontology analysis links CEBPZ, NOC2L and NOC3L depletion to rRNA processing and ribosome biogenesis in tumor cells

To explore the functional significance of CEBPZ, NOC2L and NOC3L in cancer, we analyzed data from the Cancer Dependency Portal (https://depmap.org/portal/), which utilizes CRISPR screening to identify genes that are crucial for tumor growth.

Starting from the CRISPR knockdown data from DepMap, we extracted and performed our analysis on the top 100 co-expressed genes (representing functionally correlated candidates that are likely to share pathways or regulatory mechanisms) for each of the three genes of interest (Table S2). We performed GO enrichment analysis to examine coregulated pathways and identify the functional implications of our genes and the possible contributions of individual heterodimers and the assembled complex.

The GO enrichment analysis of the shared genes reveals a network of interconnected biological processes, predominantly in ribosome biogenesis, RNA processing and rRNA maturation, which are enriched across all three data sets (Fig. 2A–C). Network visualization (Fig. 2D) from these data highlights the functional clustering of genes involved in RNA metabolism and ribosomal assembly. The network illustrates the functional interdependence of these processes, implying that perturbations in CEBPZ, NOC2L or

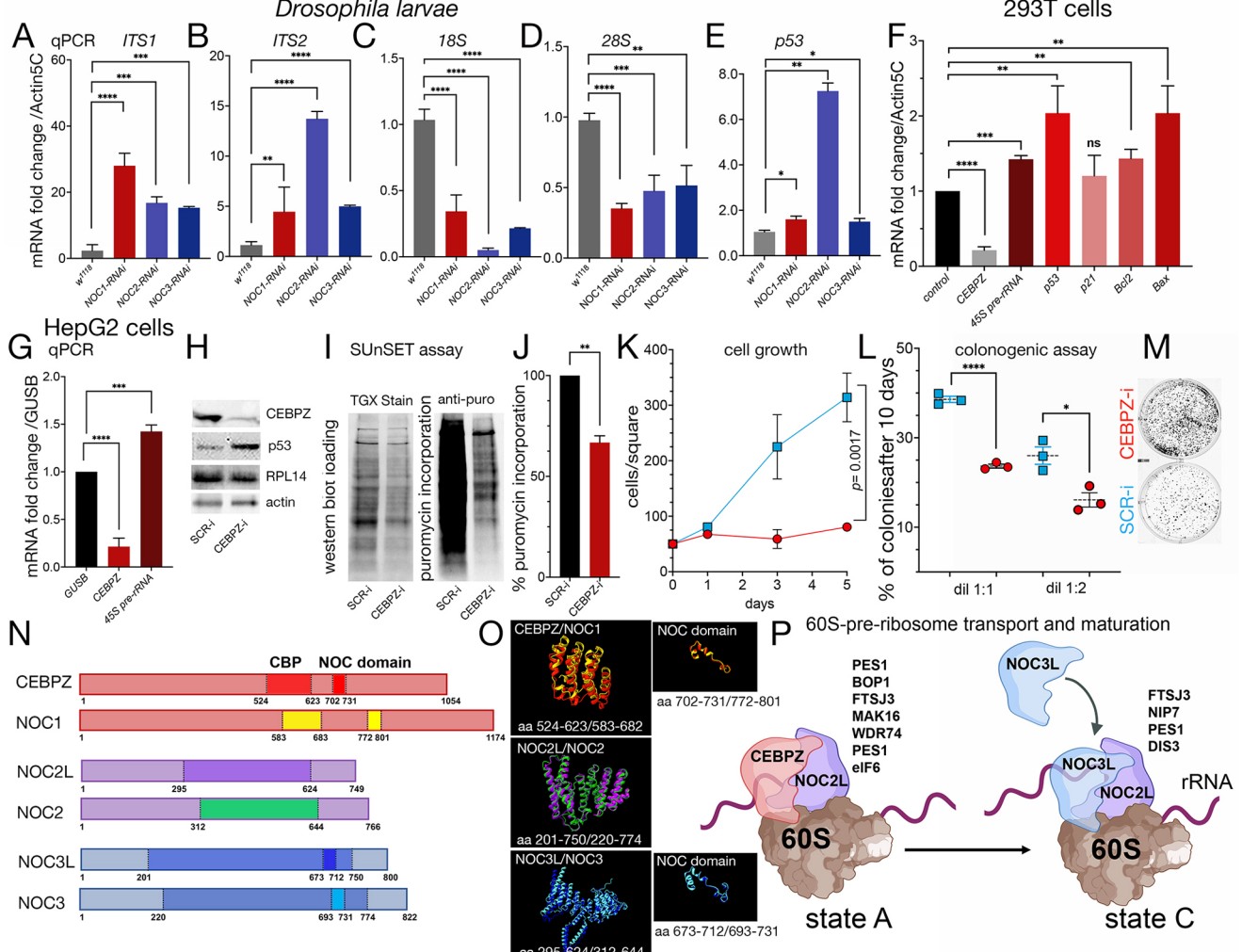

**Fig. 1. Reduction of NOC proteins disrupts rRNA processing, increases p53 expression, and reduces protein synthesis and cell growth.**
(A–E) Reduction in the levels of Nucleolar Complex proteins (NOCs) in *Drosophila* leads to rRNA accumulation, impaired maturation of *18S* and *28S* rRNA and elevated p53 levels. (A–E) qRT-PCRs from *Drosophila* whole larvae with ubiquitously reduced *NOC1, NOC2* or *NOC3* showing an accumulation of pre-rRNAs, analyzed using the internal transcribed spacers 1 and 2 (*ITS1* and *ITS3*) (A,B), and a reduction of *18S* and *28S* rRNA expression (C,D) and of *p53* mRNA (E). The level of *NOC1*, *NOC2* and *NOC3* mRNAs is calculated over control *w^1118* animals; data are expressed as fold increase relative to the control *Actin5C*; RNA interference efficiency is shown in Fig. S1. (F–M) Reducing the levels of human CEBPZ in tumor cells results in rRNA accumulation, impaired protein synthesis and decreased cell growth. (F,G) qRT-PCR analysis of HEK 293FT cells (F) and HepG2 cells (G) following siRNA-mediated silencing of CEBPZ, showing increased levels of 45S pre-rRNA and p53 mRNA. Expression levels are presented as fold change relative to control, normalized to β-glucuronidase (GUSB). Data from A–G are presented as mean±s.d. for at least three independent experiments **P<0.01; ***P<0.001; ****P<0.0001 (unpaired two-tailed Student's *t*-test). (H) Western blot from HepG2 lysates of cells transfected with a scramble siRNA (SCR-i) or for siCEBPZ (CEBPZ-i) showing the level of CEBPZ, p53, RPL14 used as an unrelated protein, and actin as a control loading. Representative of two independent experiments. (I) SUnSET assay; western blot from cells treated with 1 µg/ml of puromycin shows the relative changes in protein synthesis using anti-puromycin antibodies in cells treated with siSCR or siCEBPZ; total protein loading is shown using stain-free technologies (TGX Stain-Free Fastcast). (J) Quantification of the relative change in the rate of puromycin incorporation between SCR-i and CEBPZ-i data was analyzed from two independent experiments (mean±s.d.). (K–M) Cell growth expressed at day 1, 3 and 5 (K), and plating efficiency (PE) percentage (number of colonies formed/number of cells seeded × 100) at 10 days after treatment with siCEBPZ and siCTL (L); data are expressed from duplicate samples measured at three points (three different days) for cells originally plated at two different concentrations as mean±s.d. *P<0.05; ****P<0.0001 (unpaired two-tailed Student's *t*-test). (M) Representative photos of colonies at 10 days of treatment. (N,O) Schematic representation of *Drosophila* and human NOC1/CEBPZ, NOC2/NOC2L and NOC3/NOC3L proteins. CEBPZ contains a CBP (CCAAT binding domain in dark red) with 32% amino acid-sequence identity with *Drosophila* NOC1, and a conserved NOC domain (in red and yellow). This domain is also present in NOC3L and NOC3 (highlighted in blue and azure). NOC2 shares an overall 36% of amino acid identity between *Drosophila* and human NOC2L, with the highest identity region (47%) represented with a box (in purple and green); full sequences and alignments are shown in Table S1. (O) The predicted structural similarity of the conserved regions between CEBPZ/NOC1, NOC2L/NOC2 and NOC3L/NOC3 was obtained by a simulation from the AlphaFold database (with and without modification) and ChimeraX analysis. (P) Schematic representation of the CEBPZ–NOC2L and NOC2L–NOC3L heterodimers at states A and C of the 60S ribosomal maturation. The graph also includes proteins identified through DepMap analysis (Table 1) due to their significant correlation with the expression of the corresponding heterodimers. Their role in the ribosomal complex is illustrated according to the maturation state (A and C) of the 60S ribosome subunit as described previously (Vanden Broeck and Klinge, 2023, 2024). Fig. 1P was created in BioRender. Bellosta, P., 2025. https://BioRender.com/p57w052. This figure was sublicensed under CC-BY 4.0 terms.

Journal of Cell Science

**Table 1. Genes positively correlated with CEBPZ and NOC2L upon their reduction in tumor cells and their functional annotation from the UniProt database**

| Name | Function of the genes positively correlated with CEBPZ and NOC2L reduction |
|---|---|
| BOP1 | Component of the PeBoW complex, required for maturation of 28S and 5.8S ribosomal RNAs |
| **DDX21** | **RNA helicase acts as a sensor of the transcriptional status of RNA polymerase (Pol) I and II: promotes ribosomal RNA (rRNA) processing and transcription from polymerase II (Pol II)** |
| DDX51 | ATP-binding RNA helicase involved in the biogenesis of 60S subunits |
| **DDX56\*** | **Nucleolar RNA helicase that controls nucleolar integrity and RiBi** |
| DHX33 | Implicated in nucleolar organization, stimulates RNA polymerase I transcription of the 47S precursor rRNA. Associates with ribosomal DNA (rDNA) loci where it is involved in POLR1A recruitment |
| DIMT1 | Demethylates two adjacent adenosines in the loop of a conserved hairpin near the 3′-end of 18S rRNA in the 40S particle. Involved in the pre-rRNA processing leading to small subunit rRNA |
| DIS3‡ | Putative catalytic component of the RNA exosome complex 3′->5′ exoribonuclease activity |
| DKC1 | Small nucleolar ribonucleoprotein (H/ACA snoRNP) complex, catalyzes pseudouridylation of rRNA |
| EIF6 | Binds to the 60S subunit and prevents its association with the 40S to form the 80S |
| FTSJ3 | RNA 2′-O-methyltransferase involved in the 34S pre-rRNA to 18S rRNA and in 40S formation |
| GRWD1 | Histone binding-protein that regulates chromatin dynamics and mini chromosome maintenance |
| IMP4 | Component of the 60-80S U3 small nucleolar ribonucleoprotein (U3 snoRNP) |
| ISG20L2 | 3′-> 5′-exoribonuclease involved in ribosome biogenesis in the processing of the 12S pre-rRNA |
| LAS1L | Required for the synthesis of the 60S ribosomal subunit and maturation of the 28S rRNA |
| MAK16 | Important for the maturation of LSU-rRNA and 5.8S rRNA |
| MYBBP1A | May activate or repress transcription via interactions with sequence specific DNA-binding proteins |
| NAT10 | RNA cytidine acetyltransferase catalyzes $N_4$-acetylcytidine modification on18S and rRNA |
| NHP2 | Required for ribosome biogenesis and telomere maintenance. Part of the H/ACA small nucleolar ribonucleoprotein complex, which catalyzes pseudouridylation of rRNA |
| **NIP7** | **Required for 34S pre-rRNA processing and 60S ribosome assembly** |
| NOC4L | Nucleolar complex-associated protein 4-like protein |
| **NOL12\*** | **RNA binding protein that plays a role in RNA metabolism, the resolution of DNA stress, nucleolar organization, regulates the levels of nucleolar fibrillarin and nucleolin in pre-rRNA processing** |
| NOL9 | Involved in rRNA processing, for the processing of the 32S precursor into 5.8S and 28S rRNAs |
| NOP16‡ | Involved in the biogenesis of the 60S ribosomal subunit |
| PAK1IP1 | Negatively regulates the PAK1 kinase |
| PDCD11 | Essential for the generation of mature 18S rRNA, for cleavages at sites A0, 1 and 2 of the 47S |
| PELP1\* | Component of the PELP1 complex involved in the 28S rRNA maturation and transit of the pre-60S |
| **PES1‡** | **PeBoW complex, required for the maturation of 28S and 5.8S RNAs and formation of the 60S** |
| POLR1E | Component of RNA polymerase I polymerase which synthesizes ribosomal RNA precursor |
| POLR1G | Component of RNA polymerase I (Pol I) which synthesizes ribosomal RNA precursors |
| **POP5** | **Component of ribonuclease P, that generates mature tRNA molecules by cleaving their 5′-ends** |
| PPAN | A chimeric transcript, characterized by the first third of PPAN exon 12 joined to P2RY11 exon 2 |
| PWP1 | Regulates Pol I-mediated rRNA biogenesis and, probably, Pol III-mediated transcription |
| PWP2 | Part of the small subunit (SSU) processome, precursor of the small eukaryotic ribosomal subunit |
| RPL13A | Associated with ribosomes but is not required for canonical ribosome function |
| RPP14\* | Ribonucleoprotein complex that generates mature tRNA molecules. |
| **RPP38** | **Component of ribonuclease P complex, generates mature tRNA molecules by cleaving their 5′-ends** |
| RRP1\* | Critical role in the generation of 28S rRNA |
| RRP12 | Required for nuclear export of both pre-40S and pre-60S subunits |
| **TAF1C** | **Component of the transcription factor SL1/TIF-IB complex, which is involved in the assembly of the PIC (pre-initiation complex) during RNA polymerase I-dependent transcription** |
| TBL3 | Part of the small subunit (SSU) processome, precursor of the small eukaryotic ribosomal subunit |
| TIMM50‡ | Component of the TIM23 complex, mediates the translocation of transit peptide-containing proteins |
| URB2 | Essential for hematopoietic stem cell development through the regulation of p53/TP53 pathway |
| UTP20 | Part of the small subunit (SSU) processome, precursor of the small eukaryotic ribosomal subunit |
| UTP23 | Involved in rRNA-processing and ribosome biogenesis |
| UTP4 | Nucleolar processing of pre-18S ribosomal RNA. Part of the small subunit (SSU) processome |
| WDR18 | Component of the PELP1 complex involved in the 28S rRNA maturation and transit of the pre-60S |
| **WDR36** | **Part of the small subunit (SSU) precursor of the small eukaryotic ribosomal subunit** |
| **WDR74** | **Regulatory protein of the MTREX-exosome complex involved in the synthesis of the 60S rib.subunit** |
| WDR75 | Part of the small subunit (SSU) processome, precursor of the small eukaryotic ribosomal subunit |

\*Positively correlating when CEBPZ and NOC3 are reduced; ‡positively correlate when all three genes are reduced. The whole list of genes is available in Table S2. Bold indicates genes associated with R-loop formation and metabolism. Data are from UniProt Consortium (2024).

NOC3L can significantly impact ribosomal function and rRNA processing. From this analysis, it is plausible that decreasing the expression levels of CEBPZ or NOC2L disrupts the formation of CEBPZ–NOC2L heterodimers, leading to defects in rRNA accumulation and affecting the maturation of the ribosomal subunits. Additionally, although to a lesser extent, we found that reduction of the NOC2L–NOC3L interaction affects RNA processing and ribosome biogenesis (as inferred from GO analysis, Fig. 2B),

reinforcing our hypothesis that both pairs play a role in regulating rRNA maturation, and impairment of either heterodimer disrupts rRNA maturation, leading to its accumulation.

Table 1 presents the genes whose expression levels change in parallel with CEBPZ and NOC2L in tumor cells (i.e. positively correlated, either increasing or decreasing together), with their functional annotations from the UniProt database. Notably, as CEBPZ has previously been identified as an R-loop-associated

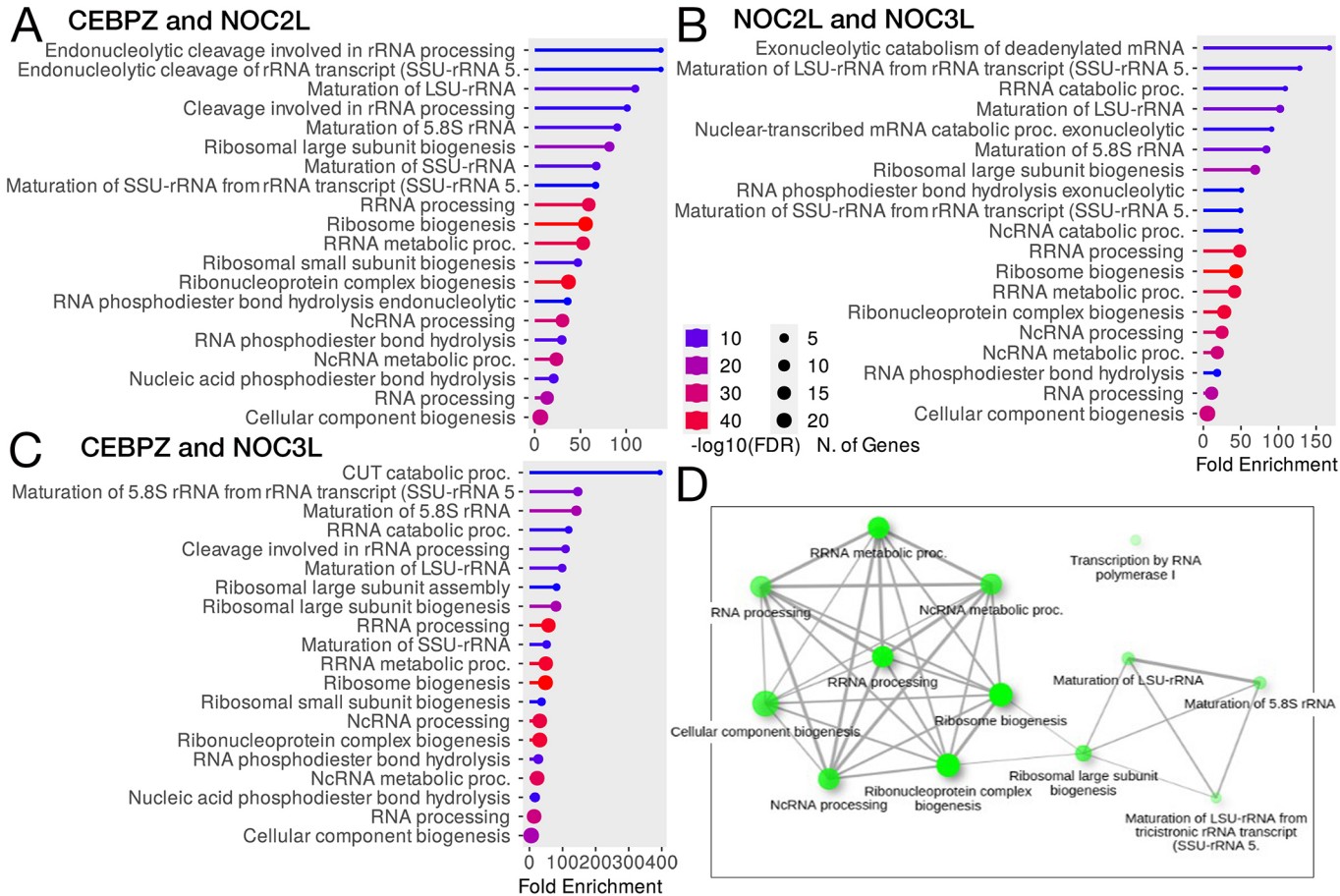

**Fig. 2. GO enrichment of the shared co-regulated genes between CEBPZ, NOC2L and NOC3L.** (A–C) GO enrichment (Biological Processes) of the shared co-regulated genes between the two indicated target genes. The results are ordered based on the fold enrichment; the circle size represents the number of genes in the pathway, the color of the bars the $-log_{10}$(FDR), and the length of the bar the enrichment. (D) GO network representation. Brighter nodes are more significantly enriched gene sets. Bigger nodes represent larger gene sets. Thicker edges represent a higher number of genes shared between the two sets.

factor (Wu et al., 2021), we also highlight in bold genes associated with R-loop metabolism (22% of the total genes).

### CEBPZ, NOC2L and NOC3L tumor expression analysis

Dysregulation of rRNA processing is a hallmark of cancers. We analyzed the expression across diverse tumor types based on the hypothesis that CEBPZ, NOC2L and NOC3L form regulatory heterodimers that modulate this process. Specifically, we examined their expression levels using TCGA data from the UALCAN cancer portal (Chandrashekar et al., 2017, 2022) (Fig. 3A–C). This analysis revealed that although all three genes were significantly upregulated ($P<0.05$) in most tumors (Fig. 3D, in red), their expression was significantly reduced ($P<0.05$) in acute myeloid leukemia (LAML), and kidney chromophobe carcinoma (KICH) (Fig. 3D, in green). In contrast, in KIRC and KIRP renal carcinomas, although the expression of CEBPZ was low, that of NOC2L and NOC3L increased (Fig. 3D). These data provide evidence of different controls for the expression of these genes in different cell or tissue contexts. Indeed, as these genes might be upregulated in many solid tumors, to support increased ribosome production and protein synthesis for rapid cellular proliferation, their selective downregulation in hematopoiesis and kidney tumors KICH might suggest a tissue-specific or cell-specific regulation. Moreover, the fact that NOC2L and NOC3L are upregulated in KIRC and KIRP might indicate that they exert a dual function, perhaps in transcriptional regulation or cell cycle control, beyond rRNA processing.

We then examined CEBPZ, NOC2L and NOC3L expression patterns across various stages of tumor development using TGCA datasets from the UALCAN portal. This analysis revealed a significant increase in their expression in liver hepatocellular carcinomas (LICH) and lung adenocarcinoma (LUAD) (Fig. 4A,B). The observed upregulation correlated with the progression of tumor stages. The expression levels of LIHC in stage 4 were comparable to those in controls, possibly because of the reduced sample size for this stage. Similar upregulation was found for rectum adenocarcinoma (READ) and stomach adenocarcinoma (STAD) (data not shown), suggesting that CEBPZ, NOC2L and NOC3L overexpression might contribute to the aggressiveness of these carcinomas, probably by sustaining the high rate of rRNA processing and ribosome activity in these tumors. Among kidney cancer subtypes, chromophobe renal cell carcinomas (KICH) showed consistently and significantly reduced expression levels of CEBPZ, NOC2L and NOC3L (Fig. 4C), whereas other subtypes, such as KIRC (clear cell) and KIRP (papillary), displayed more variable expression patterns (Fig. 4D,E). Further investigation of the co-expression dynamics of these genes in kidney cancers and Pearson correlation analysis revealed a consistently strong positive correlation between CEBPZ and NOC3L across all kidney tumor types, suggesting potential co-regulation. In contrast, CEBPZ and NOC2L exhibited weak negative correlations in KIRC and KIRP, pointing to subtype-specific regulatory differences. These distinct patterns suggest a possible dual behavior of the NOC proteins in kidney cancers – whereas CEBPZ and NOC3L might

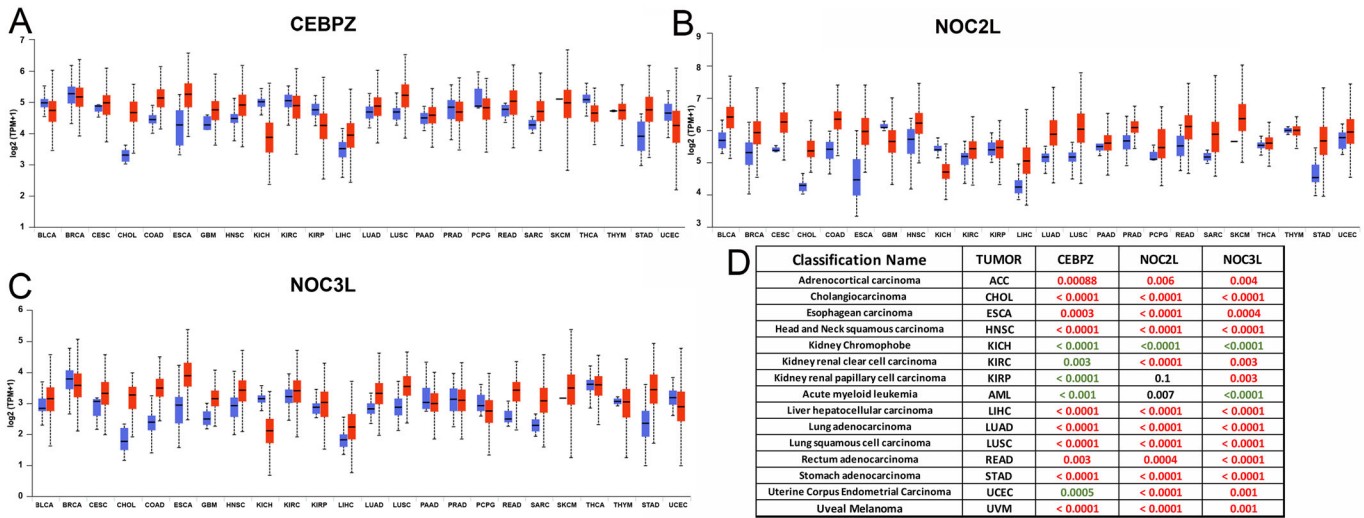

**Fig. 3. Gene expression analysis by TCGA.** (A–C) Expression levels of *CEBPZ* (A), *NOC2L* (B) and *NOC3L* (C) in tumor (red) versus normal (blue) tissues across various cancer types. The box represents the 25–75th percentiles, and the median is indicated. The whiskers show the minimum to maximum range. Outliers are excluded from the plots. (D) Summary panel highlighting cancer types where the expression of at least two of the three genes is significantly altered (*P*<0.05), as reported in the UALCAN portal. Red indicates overexpression; green indicates downregulation in tumor tissues relative to normal control.

function cooperatively, NOC2L might act independently or even antagonistically in certain contexts. Interestingly, this divergence is less pronounced in other tumor types, where all three genes tend to show more coordinated expression. These findings suggest that kidney tumors might harbor unique regulatory networks for NOC proteins, potentially influenced by tissue-specific mechanisms or the tumor microenvironment.

To deeply investigate the co-expression patterns of CEBPZ, NOC2L and NOC3L within kidney tumors, we analyzed the TCGA datasets publicly accessible through FireBrowse (Cerami et al., 2012). A Pearson correlation coefficient was calculated between the expression of the three genes (*CEBPZ*, *NOC2L* and *NOC3L*) in the three kidney carcinoma subtypes (KICH, KIRC, and KIRP). This analysis showed that CEBPZ and NOC3L exhibited the highest direct correlation in all three tumors (Fig. 5). Meanwhile, a weak negative correlation existed between *CEBPZ* and the other two genes in KIRC and KIRP. This suggests their potential distinct regulation across various kidney cancers.

### Impact of CEBPZ, NOC2L and NOC3L expression on survival for individuals with cancer

Finally, we analyzed the correlation between the expression of CEBPZ, NOC2L and NOC3L and survival for individuals with cancer to explore their potential impact on disease outcomes across tumor types. Fig. 6 displays a heatmap of hazard ratios for the genes, providing a visual summary of how their expression is related to survival for various tumors.

These data suggest that the expression of CEBPZ and NOC2L in adenoid cystic carcinoma (ACC) might be associated with poorer tumor prognosis. This means that as the expression levels of the genes increase, the severity or aggressiveness of cancer might also increase, potentially leading to worse outcomes for individuals with that cancer. A similar trend was observed in brain lower-grade glioma (LGG), liver hepatocellular carcinoma (LIHC) and lung adenocarcinoma (LUAD), with different degrees of significance (Fig. 6). In contrast, there was an inverse correlation between CEBPZ and NOC3L expression and outcomes in kidney chromophobe carcinoma. Additionally, CEBPZ, NOC2L, and NOC3L exhibited distinct expression patterns during the progression of kidney

carcinoma, with a marked decrease observed in KICH (Fig. 4C). These findings reinforce our observation that impairing RNA processing and metabolism in ribosome biogenesis, through downregulation of CEBPZ, NOC2L or NOC3L, alters key cellular pathways, potentially contributing to tumor progression in a cell-type-specific manner, particularly in kidney-derived cancers, such as KICH.

## DISCUSSION

This study reveals a conserved and novel role for the human nucleolar proteins CEBPZ, NOC2L and NOC3L in ribosome biogenesis. Our initial analysis of the NOC1 protein interactome in flies identified NOC2 and NOC3 proteins as components of the NOC1–nucleolar complex (Manara et al., 2023). These findings build on studies in yeast, where NOCs were shown to form NOC1–NOC2 and NOC2–NOC3 heterodimers necessary for rRNA processing and 60S ribosome maturation (Dorner et al., 2023; Milkereit et al., 2001).

Based on these observations, we hypothesized that in humans, CEBPZ, NOC2L, and NOC3L also function as heterodimers to control rRNA maturation as illustrated in the model in Fig. 1P. However, although NOC1 has been previously characterized in yeast as a factor primarily involved in the maturation of the 60S ribosomal subunit, our data reveal that its depletion also impairs the processing of *18S* rRNA, a component of the 40S subunit. The simultaneously reduced *18S* and *28S* rRNA maturation upon NOC1 reduction indicates a more global effect on ribosomal RNA processing and points to a central, conserved function of NOC1 in coordinating the assembly and maturation of ribosomal components. Knockdown of CEBPZ leads to pre-rRNA accumulation and activation of p53, consistent with a canonical nucleolar stress response (Lindstrom et al., 2022). This upregulation of p53 was first observed in *Drosophila* and linked to a DNA damage response upon NOC1 reduction (Pederzolli et al., 2025). Moreover, we find p53 is upregulated when each of the NOC genes was reduced (Fig. 1E), an event that might suggest that reduction of each of the three NOCs induces p53 upon ribosomal stress to activate the surveillance pathways. We propose a similar mechanism for the vertebrate homologs, CEBPZ, NOC2L and NOC3L, where the role of p53 as a downstream effector also emphasizes the potential tumor-suppressive consequences of NOC dysfunction in some tumors. The dual roles of NOC proteins in supporting tumor growth (via ribosome biogenesis)

Journal of Cell Science

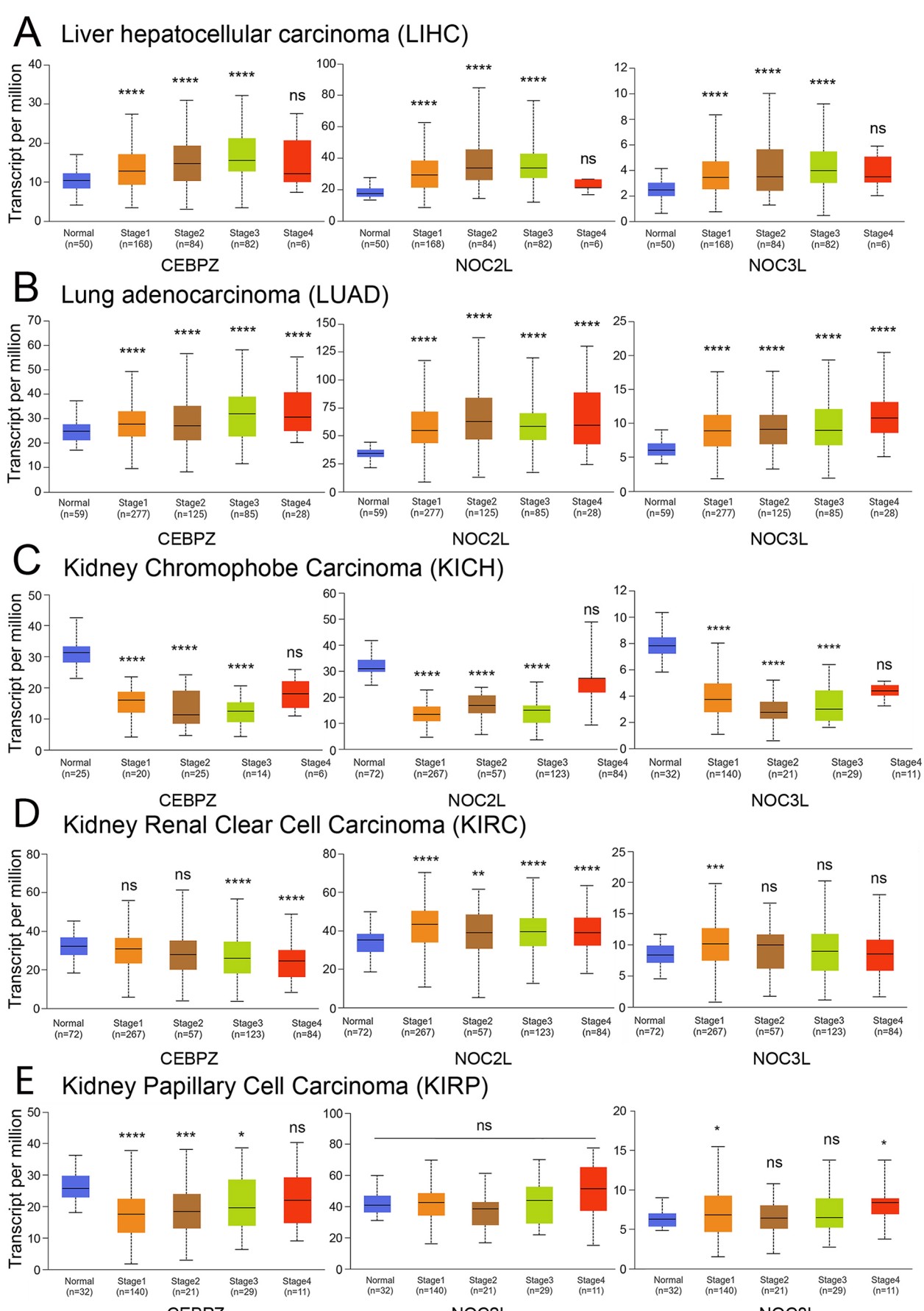

**Fig. 4.** See next page for legend.

**Fig. 4. Graphs illustrating expression levels across various cancer types during tumor progression.** Data from UALCAN highlighting the correlation between gene expression and disease progression, demonstrating significant changes in the expression of CEBPZ, NOC2L and NOC3L with advancing disease stages. (A) Liver hepatocellular carcinoma (HCC); (B) lung adenocarcinoma (LUAD); (C) kidney chromophobe carcinoma (KICH); (D) kidney renal clear cell carcinoma (KIRC); (E) kidney papillary cell carcinoma (KIRP). The statistical analysis was performed by comparing the expression levels of the indicated genes in tumors at various stages to their expression levels in normal control tissue. The box represents the 25–75th percentiles, and the median is indicated. The whiskers show the minimum to maximum range. Outliers are excluded from the plots. *$P<0.05$, **$P<0.01$, ***$P<0.001$, ****$P<0.0001$, ns: not significant (estimated by unpaired two-tailed Student's $t$-test).

and potentially triggering p53-mediated arrest when disrupted highlight their context-dependent functions. This complexity might account for the variable expression patterns observed across different cancer types and their mixed prognostic significance.

The association between the aberrant expression of NOCs and tumor progression suggests that dysregulation of NOC protein expression could contribute to oncogenesis (Fig. 4). TGCA and GO analyses show that CEBPZ, NOC2L and NOC3L expression levels correlate with genes relevant to rRNA maturation and ribosome processes. Thus, their overexpression might enhance ribosome biogenesis to sustain tumor progression (Table 1).

Differential expression of CEBPZ, NOC2L and NOC3L across cancer types with varying prognostic implications implies tissue-specific or context-dependent roles. Indeed, their elevated expression in certain cancers correlates with poor prognosis (Fig. 6). Conversely, reduced expression in other contexts might reflect a tumor-suppressive role or vulnerability in specific oncogenic settings. Paradoxically, in kidney carcinomas and AML, the reduced levels of NOC proteins might impair ribosomal function, triggering cell-type-specific apoptosis, or activate senescence programs (Nishimura et al., 2015; You and Wu, 2025), where this selective pressure could favor the emergence of aggressive clones through the accumulation of mutations that bypass these fail-safe mechanisms. A similar trend in the altered expression of other ribosome-biogenesis factors, such as dyskerin and fibrillarin, has been observed in tumorigenesis (Penzo et al., 2017; Zhang et al., 2024). Fibrillarin upregulation correlates with poor prognosis in breast cancer and AML (Luo and Kharas, 2024; Marcel et al., 2013), whereas its downregulation is also linked to adverse outcomes in breast cancer (Nguyen Van Long et al., 2022). Similarly, dyskerin overexpression has been observed in breast and prostate cancers (Kan et al., 2021; Stockert et al., 2019), whereas reduced levels are associated with breast and endometrial cancer progression (Alnafakh et al., 2021; Zacchini et al., 2022).

The essential role of *CEBPZ*, *NOC2L* and *NOC3L* genes in vertebrate development (e.g. embryonic lethality in knockout mice) and hematopoiesis (as shown in zebrafish) and the control of genes of RNA metabolism, also highlights their importance beyond basic ribosome assembly, potentially linking them to pathways that control RNA–DNA hybrid resolution or modification. Using a cross-species approach, we show these genes are crucial for rRNA maturation and overall ribosomal activity. Notably, our findings indicate that NOC family members do not functionally compensate for one another, suggesting that each protein contributes uniquely to RNA processing. This non-redundant function is crucial for cellular growth and viability in *Drosophila* (Destefanis et al., 2022) and human cells, emphasizing their fundamental and conserved role in maintaining ribosomal homeostasis. Indeed, we observed that a significant proportion (~20%) of genes correlating with CEBPZ and NOC2L expression encode for regulators or participants in R-loop metabolism (Table 1, highlighted in bold). In support of this, recent proteomic studies have identified CEBPZ as a component of protein complexes associated with R-loops (Wu et al., 2021), RNA–DNA hybrid structures that typically form during transcription. If not properly resolved, R-loops can induce RNA stress and DNA damage, contributing to genome instability (Petermann et al., 2022; Wu et al., 2021). This connection might imply context-dependent roles, where elevated expression of CEBPZ could represent a compensatory response to stress or ribosomal imbalance rather than direct oncogenic activity. However, its downregulation might indirectly promote genomic instability, a hallmark of cancer. Notably, R-loops can also form within the nucleolus due to the intense transcriptional activity of rRNA genes (Feng and Manley, 2022; Petermann et al., 2022; Wells et al., 2019).

Overall, our findings identify NOC1, NOC2, and NOC3 in *Drosophila* as regulators of rRNA processing and ribosome biogenesis. Preliminary data on the human homolog CEBPZ, along with studies of NOC2L and NOC3L expression in tumor datasets, suggest that these proteins have conserved roles in ribosome biogenesis across species. Notably, we found that their expression levels vary significantly across tumor types depending on cellular and tissue context, indicating a context-dependent role in cancer biology. In certain settings, elevated expression of these proteins might exert both tumor-suppressive effects by enforcing cell cycle checkpoints or mitigating replication stress, or, by contrast, their overexpression might support growth by reinforcing ribosomal biogenesis. Further data are required to clarify this conundrum.

Moreover, beyond their role in ribosome assembly, CEBPZ, NOC2L and NOC3L exhibit moonlighting activities, including functions in DNA replication, cell cycle regulation and possibly R-loop resolution. These additional roles might influence genomic stability and the fidelity of DNA replication by impacting the timing

| KICH | | | | KIRC | | | | KIRP | | | |
|---|---|---|---|---|---|---|---|---|---|---|---|
| r | | | | r | | | | r | | | |
| | CEBPZ | NOC2L | NOC3L | | CEBPZ | NOC2L | NOC3L | | CEBPZ | NOC2L | NOC3L |
| CEBPZ | 1 | −0.035481 | 0.7768214 | CEBPZ | 1 | −0.263372 | 0.5205663 | CEBPZ | 1 | −0.223414 | 0.4592963 |
| NOC2L | −0.035481 | 1 | 0.1116638 | NOC2L | −0.263372 | 1 | −0.225758 | NOC2L | −0.223414 | 1 | −0.151207 |
| NOC3L | 0.7768214 | 0.1116638 | 1 | NOC3L | 0.5205663 | −0.225758 | 1 | NOC3L | 0.4592963 | −0.151207 | 1 |
| P | | | | P | | | | P | | | |
| | CEBPZ | NOC2L | NOC3L | | CEBPZ | NOC2L | NOC3L | | CEBPZ | NOC2L | NOC3L |
| CEBPZ | NA | 0.7384625 | 0 | CEBPZ | NA | $4.50×10^{-11}$ | 0 | CEBPZ | NA | $5.10×10^{-5}$ | 0 |
| NOC2L | 0.7384625 | NA | 0.2919838 | NOC2L | $4.50×10^{-11}$ | NA | $1.92×10^{-8}$ | NOC2L | $5.10×10^{-5}$ | NA | 0.0064759 |
| NOC3L | 0 | 0.2919838 | NA | NOC3L | 0 | $1.92×10^{-8}$ | NA | NOC3L | 0 | 0.0064759 | NA |

**Fig. 5. A matrix highlighting the significant correlation between CEBPZ, NOC2L and NOC3L expression in KICH, KIRC and KIRP kidney carcinomas.** For each tumor type, the $r$ (correlation coefficient) and the $P$-value, highlighted in blue, are significant when there is a correlation higher than ±0.4. A higher absolute number means a stronger correlation between the expression levels for the samples (e.g. in KICH, when CEBPZ is low, NOC3L is also low). The lower part of the table shows the $P$-value for each correlation (calculated with a two-tailed Pearson correlation test), highlighting the significant ones ($P<0.05$).

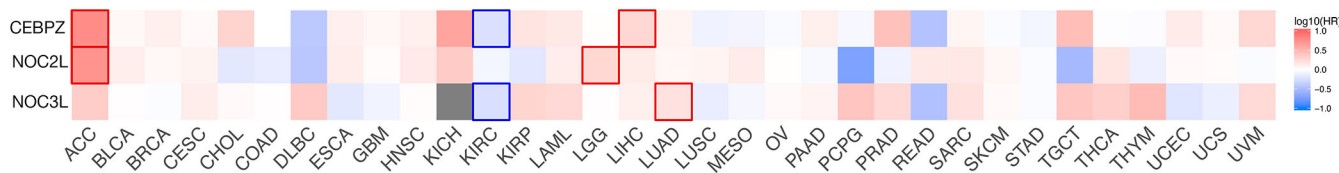

**Fig. 6. Hazard ratio heat map for CEBPZ, NOC2L and NOC3L in different tumor types.** Hazard ratio (HR) heat map for CEBPZ (ID: ENSG00000115816.13), NOC2L (ID: ENSG00000188976.10) and NOC3L (ID: ENSG00000173145.11) in the tumors listed. The median is selected as a threshold for separating high-expression and low-expression cohorts. Bounding boxes depict changes that are statistically significant ($P<0.05$). Colors from white to red indicate an HR above 1, suggesting that high gene expression is associated with a worse prognosis. Colors from white to blue indicate an HR below 1, suggesting that high gene expression is associated with a better prognosis.

of cell cycle progression. A deeper understanding of how these multifunctional activities intersect with their expression patterns in cancer underscores the need for further investigation into their potential roles in tumor development.

## MATERIALS AND METHODS
### Fly husbandry
Fly cultures and crosses were raised at 25°C on a standard medium containing 9 g/l agar (ZN5 B and V), 75 g/l corn flour, 60 g/l white sugar, 30 g/l brewers' yeast (Thermo Fisher Scientific), 50 g/l fresh yeast, 50 ml/l molasses (Naturitas), along with nipagin, and propionic acid (Thermo Fisher Scientific). The lines used, *NOC1-RNAi* (B25992), *NOC2-RNAi* (B50907) and *NOC3-RNAi* (B61872), were obtained from the Bloomington *Drosophila* Stock Center.

### Western blot and SUnSET assay
At 48 h after transfection with the relative siRNAs, HepG2 cells were incubated in medium containing 10% FBS (Sigma-Aldrich) and puromycin at 1 µg/ml (Invitrogen, Thermo Fisher Scientific) for 40 min at room temperature, then recovered in 10% serum in medium without puromycin for 30 min at room temperature. Cells were lysed in RIPA buffer supplemented with protease and phosphatase inhibitors. Protein concentrations were determined using the BCA assay (Thermo Fisher Scientific). Equal amounts of protein (40 µg) were separated by 10% SDS-PAGE and blotted using anti-puromycin antibodies (1:5000, 12D10, Sigma-Aldrich). Total protein loading was analyzed using stain-free technologies (TGX Stain-Free Fastcast). Signal detection was performed using an ECL substrate (GE Healthcare) and visualized using a ChemiDoc imaging system (Bio-Rad). Other antibodies used were anti-p53 (1:1000, clone BP53-12, Novocastra), RPL14, and mouse anti-β-actin (1:4000, clone AC-74, Sigma-Aldrich).

### Cell culture and RNAi treatment
HEK-293FT and HepG2 cells (both from the cellular facility at CIBIO) were cultured in DMEM (Corning Inc.) supplemented with 10% FBS, 2 mM L-glutamine, 100 U/ml penicillin, and 100 µg/ml streptomycin (all from Sigma-Aldrich) and maintained at 37°C, 5% $CO_2$ in a humidified incubator. For RNA interference, cells were transfected with Lipofectamine RNAiMAX (Thermo Fisher Scientific) following the manufacturer's specifications. Transfected siRNA sequences were as follows: siRNA CEBPZ 1: 5′-rCrArArArArGrUrCrArGrUrArCrUrArArArArArArArGr-CAA-3′; siRNA CEBPZ 2: 5′-rUrUrGrCrUrUrUrUrUrArGrUrArCr-UrGrArCrUrUrGrArG-3′; negative control: scrambled negative control (siSCR) DsiRNA (all from Integrated DNA Technologies). Cells were harvested at 72 h after transfection, total RNA was extracted.

### Cell growth and the clonogenic assay
HepG2 cells were plated in six-well plates after 48 h of transfection at the same concentration of 800 cells/ml and 400 cells/ml, for the siSCR and siCEBPZ in duplicates. Cells were fixed and stained on days 1, 3 and 5 with a methanol and Trypan Blue solution (Sigma-Aldrich) for 30 min, then washed with water to remove excess staining. Photos were taken using a Zeiss Axio Imager M2 microscope. Four different frames/areas from each well were taken, and cells were counted from photos using ImageJ and data

graphed using GraphPad Prism v10. Clonogenic efficiency was calculated based on the initial number of cells plated at day 0, then counted at days 1, 3 and 5 (Misra and Rajawat, 2021).

### RNA extraction and RT-PCR
RNA was extracted from *Drosophila* larvae or HEK 293FT and HepG2 human cancer cells using the RNeasy Mini Kit (Qiagen) following the manufacturer's instructions. The isolated RNA was quantified using a Nanodrop2000. Total RNA (1000 ng) was reverse transcribed into complementary DNA (cDNA) using SuperScript IV VILO Master Mix (Invitrogen). The obtained cDNA was used for quantitative real-time PCR (qRT-PCR) with the SYBR Green PCR Kit (Qiagen). The assays were performed on a Bio-Rad CFX96 machine and analyzed using Bio-Rad CFX Manager software. Transcript abundance was normalized using *Actin5c*. The primer list was published in Destefanis et al. (2022). For human targets, primers were purchased from Integrated DNA Technologies: CEBPZ F: 5′-TCTCATCCAAAGTAGCCAGCAT-3′, R: 5′-TCTCATCCAAAG-TAGCCAGCAT-3′; 45S pre-rRNA F: GAACGGTGGTGTGTCGTTC-3′, R: 5′-GCGTCTCGTCTCGTCTCACT-3′; p21 F: 5′-TGGGGATGTCCGT-CAGAACC, R: 5′-TGGAGTGGTAGAAATCTCTCATGCT-3′; BCL2 F: 5′-ATCGCCCTGTGGATGACTGAGT, R: 5′-GCCAGGAGAAATCAA-ACAGAGGC-3′; BAX F: 5′-ATGTTTTCTGACGGCAACTTC, R: 5′-ATCAGTTCCGGCACCTTG-3′, and the GUSB housekeeping expression kit was purchased from Applied Biosystems (ref 4326320E).

### Datasets used in this study
The data supporting Fig. 2 and Table S2 are publicly available in the DepMap portal (https://depmap.org/portal). Gene-level information can be accessed using the following queries: CEBPZ (https://depmap.org/portal/gene/CEBPZ), NOC2L (https://depmap.org/portal/gene/NOC2L) and NOC3L (https://depmap.org/portal/gene/NOC3L); top 100 co-dependency datasets for each gene were used in this study.

The data underlying Figures 3 and 4 are available from UALCAN (https://ualcan.path.uab.edu/openly). Gene expression profiles of CEBPZ, NOC2L and NOC3L can be retrieved through the UALCAN analysis portal (https://ualcan.path.uab.edu/cgi-bin/ualcan-res.pl). Additional tumor expression data for these genes were obtained from GEPIA2 (http://gepia2.cancer-pku.cn).

### Correlations
TCGA data were accessed through FireBrowse (Broad Institute TCGA Genome Data Analysis Center, 2016; Firehose 2016-01-28 run. Broad Institute of MIT and Harvard; doi:10.7908/C11G0KM9) and processed in-house using the r-corr function of the Hmisc package in R software (https://www.rdocumentation.org/packages/Hmisc/versions/5.1-3).

### Differential expression and HR heat-map
Differential expression analysis was conducted using the GEPIA2 (Tang et al., 2019) platform to compare gene expression profiles between tumor and non-tumor samples derived from the TCGA and GTEx datasets. GEPIA2 was also used to produce a survival heat map of the hazard ratio.

### Gene ontology enrichment
GO and KEGG enrichment analyses were performed using ShinyGO (Ge et al., 2020) with a false discover rate (FDR) cutoff of 0.5. The shared co-regulated genes between CEBPZ, NOC2L and NOC3L were tested.

## Co-expressed genes

Broad DepMap (https://depmap.org/portal/; Project data public release 24Q4) was used to determine the gene dependencies after CRISPR in cancer cell lines (Arafeh et al., 2025; Fong et al., 2025; Meyers et al., 2017; Tsherniak et al., 2017). The Broad DepMap project reports essentiality scores using the Chronos algorithm (Dempster et al., 2021). A lower score indicates a greater probability that the gene of interest is essential in a specific cell line. A score of 0 denotes a non-essential gene, whereas a score of −1 reflects the median for pan-essential genes.

## Tumor stage expression

The expression levels of the three genes of interest have been explored through UALCAN at https://ualcan.path.uab.edu.

## AI use declaration

Portions of the text were prepared with the assistance of ChatGPT (OpenAI) to improve clarity and language. After using these services, the authors reviewed and edited the content as needed and take full responsibility for the content of the publication.

## Statistical analysis

Unpaired two-tailed Student's $t$-test analysis and one-way ANOVA with and Tukey's multiple comparisons tests were performed using GraphPad-PRISM8. $P$-values are indicated with asterisks: $*P<0.05$, $**P<0.01$, $***P<0.001$, $****P<0.0001$, respectively.

## Acknowledgements

We thank the Bloomington Stock Center (NIH P40OD018537). Department CIBIO Core Facilities is supported by the European Regional Development Fund (FESR) 2021–2027. This work has been supported by the Imaging Facility at CIBIO and by the initiative "Dipartimenti di Eccellenza 2023-2027 (Legge 232/2016)" funded by the MUR. This article is based on work from COST Action CA21154 TRANSLACORE, supported by COST (European Cooperation in Science and Technology) and by the Dipartimento di Eccellenza 2023-2027, Legge 232/2016 project no. 40613, funded by the MUR.

## Competing interests

The authors declare no competing or financial interests.

## Author contributions

Conceptualization: G.B., M.P., P.B.; Data curation: G.R., V.M., A.V.C., M.P., P.B.; Formal analysis: G.R., A.V.C., P.B.; Funding acquisition: P.B.; Investigation: V.M., M.P.; Supervision: P.B.; Validation: G.R., M.P.; Writing – original draft: P.B.; Writing – review & editing: G.R., V.M., A.V.C., G.B., M.P., P.B.

## Funding

This work was supported by a National Institutes of Health (NIH) Public Health Service grant from NIH-SC1DK085047 to P.B. and Pallotti Legacy for Cancer Research to M.P. G.R. is a recipient of a PhD fellowship in the PON program 'Research and Innovation' 2014-2020 with financial support from REACT-EU resources. Open Access funding provided by University of Trento. Deposited in PMC for immediate release.

## Data and resource availability

All relevant data can be found within the article and its supplementary information.

## First Person

This article has an associated First Person interview with the first author of the paper.

## Peer review history

The peer review history is available online at https://journals.biologists.com/jcs/lookup/doi/10.1242/jcs.264096.reviewer-comments.pdf

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
