## [Peer Review File · Journal of Cell Science]

***Drosophila* and human cell studies reveal a conserved role for CEBPZ, NOC2L and NOC3L in rRNA processing and tumorigenesis**

Guglielmo Rambaldelli, Valeria Manara, Andrea Vutera Cuda, Giovanni Bertalot, Marianna Penzo and Paola Bellosta

DOI: 10.1242/jcs.264096

Editor: Megan King

Review timeline

Original submission: 21 April 2025

Editorial decision: 9 July 2025

First revision received: 11 July 2025

Accepted: 20 July 2025

Original submission

First decision letter

MS ID#: jcs.264096

MS TITLE: From Flies to Humans: A Conserved Role of CEBPZ, NOC2L, and NOC3L in rRNA Processing and Tumorigenesis.

AUTHORS: Guglielmo Rambaldelli; Valeria Manara; Andrea Vutera Cuda; Giovanni Bertalot; Marianna Penzo; Paola Bellosta

ARTICLE TYPE: Research Article

Dear Dr Bellosta,

We apologize for the delay in sharing the outcome of the review of your manuscript. Obtaining expert reviewers and receiving their reports took us more time than we would have liked. Nonetheless we have now reached a decision on your manuscript.

To see the reviewers' reports and a copy of this decision letter, please go to:

As you will see, the reviewers gave favourable reports but raised some critical points that will require amendments to your manuscript. I hope that you will be able to carry these out because I would like to be able to accept your paper, depending on further comments from reviewers.

Reviewer 1

Previous elegant work published in the Journal of Cell Science (Destefanis et al 2022) showed that the nucleolar NOC proteins have a conserved role in rRNA processing, ribosome maturation, protein synthesis and animal development, building on prior work in yeast. In the current study, the authors show that depletion of the NOC1 homolog CEBPZ in human cells results in similar defects in ribosome biogenesis. The study then reports a series of correlations between NOC protein expression and factors regulating rRNA processing and ribosome maturation. Correlations are then proposed between NOC protein expression and tumor growth through data mining from TCGA datasets across different tumor types and stages.

Unfortunately, my enthusiasm for the paper is reduced due to prior published work by the authors and others in the field together with the lack of supporting experimental data to validate the ideas

drawn from the bioinformatics and data mining analysis. My concern is that the work does not significantly extend our understanding of NOC protein function beyond showing conservation of its known role in rRNA processing in human cells. The gene ontology, tumour expression and hazard ratio analyses are of interest but correlative and somewhat preliminary. No experimental data are included to support the interesting idea that CEBPZ is involved in R-loop formation. No experimental data are provided to support the claim in the title that the NOC proteins have a conserved role in tumorigenesis.

Reviewer 2

In this manuscript, authors show that NOC1, NOC2, and NOC3 are conserved nucleolar proteins essential for rRNA processing. Also, they show that their reduction impairs rRNA maturation, decreases protein synthesis, and induces cell death. In addition, authors shows that CEBPZ, NOC2L, and NOC3L are expressed in tumors in a context-dependent manner, suggesting dual roles in cancer progression and suppression.

The manuscript is an interesting and comprehensive piece of work. It is well-written, logically presented, and includes the appropriate controls. Thus, this work seems suitable for publication in the Journal of Cell Science. Nevertheless, there some minor points that should be addressed before publication.

Minor concerns:

Figure 1E: Please, include the RT-qPCR data of p53 mRNA expression under reduction of CEBPZ, NOC2L, or NOC3L, as these data is not shown.

Figure 1P: Please, change "DIS1" to "DIS3"

Page 11: Please, include a more informative explanation of results shown in Figures 4 D-E.

First revision

Author response to reviewers' comments

Reviewer 1: Previous elegant work published in the Journal of Cell Science (Destefanis et al 2022) showed that the nucleolar NOC proteins have a conserved role in rRNA processing, ribosome maturation, protein synthesis and animal development, building on prior work in yeast. In the current study, the authors show that depletion of the NOC1 homolog CEBPZ in human cells results in similar defects in ribosome biogenesis. The study then reports a series of correlations between NOC protein expression and factors regulating rRNA processing and ribosome maturation. Correlations are then proposed between NOC protein expression and tumor growth through data mining from TCGA datasets across different tumor types and stages. Unfortunately, my enthusiasm for the paper is reduced due to prior published work by the authors and others in the field together with the lack of supporting experimental data to validate the ideas drawn from the bioinformatics and data mining analysis. My concern is that the work does not significantly extend our understanding of NOC protein function beyond showing conservation of its known role in rRNA processing in human cells. The gene ontology, tumour expression and hazard ratio analyses are of interest but correlative and somewhat preliminary. No experimental data are included to support the interesting idea that CEBPZ is involved in R- loop formation. No experimental data are provided to support the claim in the title that the NOC proteins have a conserved role in tumorigenesis.

We thank the reviewer for their thoughtful comments and for highlighting our previous work. We agree with the reviewer that some of the correlations presented in this study are preliminary; however, we believe the correlative analysis is solid and provides a strong foundation for our conclusions and future work. Specifically, we have applied rigorous bioinformatic pipelines and statistical controls to ensure the robustness of the correlations between human NOCs protein

expression and tumor-related parameters.

Regarding the conservation of the mechanism, we respectfully point out that while previous studies, including our own, have demonstrated the role of NOC proteins in rRNA processing in yeast and *Drosophila*, our current data in human cells provide new insights into this role in vertebrates, generating preliminary evidence that supports the idea that the identified mechanism in flies may be evolutionarily conserved.

Even if these data are preliminary, we believe the results presented here are valuable and worth sharing with the community. We acknowledge that additional experimental validation, especially concerning the role of CEBPZ in R-loop formation and tumorigenesis, is needed, and we are currently pursuing these investigations to generate direct experimental data using different human wild-type and cancer cell lines. However, these efforts require time, and we consider the current study a significant step forward necessary to begin our understanding of the broader implications of NOC protein function.

Reviewer 2: In this manuscript, authors show that NOC1, NOC2, and NOC3 are conserved nucleolar proteins essential for rRNA processing. Also, they show that their reduction impairs rRNA maturation, decreases protein synthesis, and induces cell death. In addition, authors show that CEBPZ, NOC2L, and NOC3L are expressed in tumors in a context-dependent manner, suggesting dual roles in cancer progression and suppression.

The manuscript is an interesting and comprehensive piece of work. It is well-written, logically presented, and includes the appropriate controls. Thus, this work seems suitable for publication in the Journal of Cell Science. Nevertheless, there are some minor points that should be addressed before publication.

Minor concerns:

Figure 1E: Please, include the RT-qPCR data of p53 mRNA expression under reduction of CEBPZ, NOC2L, or NOC3L, as these data is not shown.

- We apologize for the oversight. The RT-qPCR data for p53 mRNA were mistakenly hidden in the original figure file. We have now corrected this and included the data in the revised Figure 1E.

Figure 1P: Please, change "DIS1" to "DIS3"

- We correct the typo

Page 11: Please, include a more informative explanation of results shown in Figures 4D-E.

We thank the reviewer for the suggestion. We have now expanded the results section to provide a more detailed explanation of the analysis. Adding this sentence:

Among kidney cancer subtypes, chromophobe renal cell carcinomas (KICH) showed consistently and significantly reduced expression levels of CEBPZ, NOC2L, and NOC3L (Figure 4C), while other subtypes, such as KIRC (clear cell) and KIRP (papillary), displayed more variable expression patterns (Figure 4D-E). To further investigate the co-expression dynamics of these genes in kidney cancers, we analyzed TCGA datasets from KICH, KIRC, and KIRP. Pearson correlation analysis revealed a consistently strong positive correlation between CEBPZ and NOC3L across all kidney tumor types, suggesting potential co-regulation. In contrast, CEBPZ and NOC2L exhibited weak negative correlations in KIRC and KIRP, pointing to subtype-specific regulatory differences. These distinct patterns suggest a possible dual behavior of the NOC proteins in kidney cancers: while CEBPZ and NOC3L may function cooperatively, NOC2L could act independently or even antagonistically in certain contexts. Interestingly, this divergence is less pronounced in other tumor types, where all three genes tend to show more coordinated expression. These findings suggest that kidney tumors may harbor unique regulatory networks for NOC proteins, potentially influenced by tissue-specific mechanisms or the tumor microenvironment.

Reviewer 3: SUMMARY OF THE ADVANCE MADE IN THIS PAPER AND ITS POTENTIAL SIGNIFICANCE TO THE FIELD

This manuscript describes the roles of CEBPZ, NOC2L and NOC3L in rRNA processing in fly and human cells, and the expression of these proteins in tumorigenesis. Reduction of the CEBPZ homolog of NOC1 leads to the accumulation of unprocessed 45S pre-rRNA, a reduction in protein synthesis, and inhibition of cell growth. Gene Ontology and bioinformatic analyses of CEBPZ, NOC2L and NOC3L in tumors highlight a significant correlation between their expression. The data suggest that overexpression of these proteins may promote tumor growth.

SUGGESTIONS TO AUTHORS

Major comments [Please request additional experiments only if they are essential for supporting the conclusions; authors should be encouraged to highlight any claims that are preliminary or speculative, or to discuss any pitfalls or alternative interpretations in a 'Limitations' section]

Minor comments

1. Fig. 1E (*p53* levels) is blank: no graphs or any data.

- We apologize for the oversight. The RT-qPCR data for *p53* mRNA were mistakenly hidden in the original figure file. We have now corrected this and included the data in the revised Figure 1E.

2. Should cite the first paper (Cell, 2001) reporting Noc1-3 as ribosome biogenesis proteins, and the first paper (Cell, 2002) that identified Noc3 as a replication-initiation protein.

-We thank the reviewer for this observation.

I believe the referee is indicating the Cell 2001 paper Milkereit, P., Gadal, O., Podtelejnikov, A., Trumtel, S., Gas, N., Petfalski, E., Tollervey, D., Mann, M., Hurt, E., and Tschochner, H. (2001). Maturation and intranuclear transport of pre-ribosomes requires Noc proteins. *Cell* 105, 499-509, which was included in the publication.

Instead, we add a paragraph on page 4 to highlight the original paper in yeast that described the conserved function of Noc3, in the control of DNA initiation complex from Zang, Y., Yu, Z., Fu, X., and Liang, C. (2002). Noc3p, a bHLH protein, plays an integral role in the initiation of DNA replication in budding yeast. *Cell* 109, 849-60. which we have missed. Please let me know otherwise.

3. The words in the X-axis and Y-axis of Figs. 1, 3 and 4 are too small.

-We thank the reviewer for the observation. To improve readability, we have enlarged the font size of the X- and Y-axis labels in Figures 1, 3, and 4.

4. "In conclusion, we have identified CEBPZ, NOC2L, and NOC3L as novel regulators of rRNA processing and ribosome biogenesis."

Should this statement be limited to fly?

Note that the roles of these proteins have been discovered in other organisms.

- We agree with the reviewer that this statement should be more precise. In the revised text, we now clarify that our functional studies identify CEBPZ, NOC2L, and NOC3L as regulators of rRNA processing and ribosome biogenesis in *Drosophila*. While similar roles have been described in other organisms, our data suggest that these mechanisms may be evolutionarily conserved, and preliminary evidence supports their relevance in human cells as well.

5. The title, "From Flies to Humans..." may need correction, as the study only concerns fly and human cells.

- The title has been changed to:

***Drosophila* and Human Cell Studies Reveal a Conserved Role for CEBPZ, NOC2L, and NOC3L in rRNA Processing and Tumorigenesis**

6. In the conclusion and discussion, the authors attributed the association of NOC1-3 over expression in cancer to ribosome biogenesis. However, these proteins are also involved in DNA replication and the cell cycle. Therefore, the contribution of over expression of these proteins to cancer may also be related to enhanced DNA replication and cell cycle progression. Should the discussion be revised?

- Thank you for this insightful comment. We agree that the functions of CEBPZ, NOC2L, and NOC3L extend beyond ribosome biogenesis. These proteins indeed exhibit moonlighting activities, including roles in DNA replication and cell cycle regulation, which can impact cancer biology in complex ways. Overexpression of these proteins may not solely control oncogenesis through enhanced ribosome production but could also influence genomic stability, via R-loops stability, and cell cycle checkpoints, potentially exerting tumor-suppressive effects by restraining uncontrolled proliferation or mitigating replication stress. We acknowledge that the relationship between their expression and cancer progression is likely more nuanced than initially described. Accordingly, we have revised the discussion to incorporate these additional functions and the possibility of tumor-suppressive roles, highlighting the need for further investigation into the diverse activities of CEBPZ, NOC2L, and NOC3L in cancer, with the sentence below in the discussion.

We change this last sentence in the text to:

Overall, our findings identify NOC1, NOC2, and NOC3 in *Drosophila* as regulators of rRNA processing and ribosome biogenesis. Preliminary data on the human homolog CEBPZ, along with studies of NOC2L and NOC3L expression in tumor datasets, suggest that these proteins have conserved roles in ribosome biogenesis across species. Notably, we found that their expression levels vary significantly across tumor types depending on cellular and tissue context, indicating a context-dependent role in cancer biology. In certain settings, elevated expression of these proteins may exert both tumor-suppressive effects by enforcing cell cycle checkpoints or mitigating replication stress, or, on the contrary, their overexpression supports growth by reinforcing ribosomal biogenesis. Further data are required to clarify this conundrum.

Moreover, beyond their role in ribosome assembly, CEBPZ, NOC2L, and NOC3L exhibit moonlighting activities, including functions in DNA replication, cell cycle regulation, and possibly R-loop resolution. These additional roles may influence genomic stability and the fidelity of DNA replication by impacting the timing of cell cycle progression. A deeper understanding of how these multifunctional activities intersect with their expression patterns in cancer underscores the need for further investigation into their potential roles in tumor development.

Second decision letter

MS ID#: jcs.264096R1

MS TITLE: *Drosophila* and Human Cell Studies Reveal a Conserved Role for CEBPZ, NOC2L, and NOC3L in rRNA Processing and Tumorigenesis

AUTHORS: Guglielmo Rambaldelli; Valeria Manara; Andrea Vutera Cuda; Giovanni Bertalot; Marianna Penzo; Paola Bellosta
Article Type: Research Article

Dear Dr Bellosta,

I am happy to tell you that your manuscript has been accepted for publication in Journal of Cell Science, pending standard publication integrity checks.